**Data Availability Statement:** All relevant data are within the manuscript. The tables included in the

# Effects of low dose computed tomography (LDCT) on lung cancer screening on incidence and mortality in regions with high tuberculosis prevalence: A systematic review

Debora Castanheira Pires[1]*, Luisa Arueira Chaves[2], Carlos Henrique Dantas Cardoso[3], Lara Vinhal Faria[3], Silvio Rodrigues Campos[3], Mario Jorge Sobreira da Silva[4], Tayna Sequeira Valerio[4], Mônica Rodrigues Campos[5], Isabel Cristina Martins Emmerick[6]

1 Laboratório de Pesquisa Clínica em DST e AIDS do Instituto Nacional de Infectologia Evandro Chagas, Fundação Oswaldo Cruz, Rio de Janeiro, Rio de Janeiro, Brazil, 2 Instituto de Ciências Farmacêuticas, Universidade Federal do Rio de Janeiro, Macaé, Rio de Janeiro, Brazil, 3 Departamento de Administração e Planejamento em Saúde–Escola Nacional de Saúde Pública, Fundação Oswaldo Cruz, Rio de Janeiro, Rio de Janeiro, Brazil, 4 Divisão de Ensino, Instituto Nacional de Câncer, Rio de Janeiro, Rio de Janeiro, Brazil, 5 Departamento de Ciências Sociais–Escola Nacional de Saúde Pública, Fundação Oswaldo Cruz, Rio de Janeiro, Rio de Janeiro, Brazil, 6 Division of Thoracic Surgery, Department of Surgery, UMass Chan Medical School, Worcester, Massachusetts, United States of America

* isabel.emmerick@umassmed.edu

## Abstract

### Background

Lung cancer screening (LCS) using low-dose computed tomography (LDCT) is a strategy for early-stage diagnosis. The implementation of LDCT screening in countries with a high prevalence/incidence of tuberculosis (TB) is controversial. This systematic review and meta-analysis aim to identify whether LCS using LDCT increases early-stage diagnosis and decreases mortality, as well as the false-positive rate, in regions with a high prevalence of TB.

### Methods/Design

Studies were identified by searching BVS, PUBMED, EMBASE, and SCOPUS. RCT and cohort studies (CS) that show the effects of LDCT in LC screening on mortality and secondary outcomes were eligible. Two independent reviewers evaluated eligibility and a third judged disagreements. We used the Systematic Review Data Repository (SRDR+) to extract the metadata and record decisions. The analyses were stratified by study design and incidence of TB. We used the Cochrane "Risk of bias" assessment tool.

### Results

The Preferred Reporting Items for Systematic Reviews and Meta-Analysis (PRISMA) were used. Thirty-seven papers were included, referring to 22 studies (10 RCTs and 12 cohorts). Few studies were from regions with a high incidence of TB (One RCT and four cohorts).

manuscript contain the references for the manuscripts included in the Systematic Literature Review.

**Funding:** This research was funded partially by departmental funds of the Division of Thoracic Surgery Department of Surgery - UMass Chan Medical School, Worcester – MA. Fundação de Amparo a Pesquisa do Estado do Rio de Janeiro Grant Number: E-26/ 210.131/2022. Oswaldo Cruz Foundation – Fiocruz Brazil – INOVA Grant Number: VPPCB-007-FIO-18-2-128. The funding agencies did NOT have any role in the design and conduct of the study, collection, management, analysis, and interpretation of the data; preparation, review, or approval of the manuscript; and the decision to submit the manuscript for publication.

**Competing interests:** The authors have declared that no competing interests exist.

Nonetheless, the evidence is compatible with European and USA studies. RCTs and CS also had consistent results. There is an increase in early-stage (I-II) diagnoses and reduced LC mortality in the LCDT arm compared to the control. Although false-positive rates varied, they stayed within the 20 to 30% range.

## Discussion

This is the first meta-analysis of LDCT for LCS focused on its benefits in regions with an increased incidence/prevalence of TB. Although the specificity of Lung-RADS was higher in participants without TB sequelae than in those with TB sequelae, our findings point out that the difference does not invalidate implementing LDCT LCS in these regions.

## Trial registration

**Systematic review registration** Systematic review registration PROSPERO CRD42022309581.

## Introduction

Lung cancer is the second most common malignancy, responsible for 2.21 million new cases and the first in deaths, leading to 1.80 million deaths worldwide. About 12% of all new cancers are lung cancer [1]. In most countries, age-standardized 5-year net survival was 10–20%, the highest rate in Japan (32.9%). The best rates were in 12 countries (range 20–30%): Mauritius, Canada, and the USA; four Asian countries (China, Korea, Taiwan, and Israel); and five European countries (Latvia, Iceland, Sweden, Austria and Switzerland) [2]. Tobacco use is the main risk factor for developing lung cancer [3], causing 63% of global deaths and more than 90% in countries where smoking is prevalent [4]. Governments are implementing tobacco control policies as a primary strategy for preventing lung cancer [5].

Due to the low survival rates when diagnosed with advanced-stage and specific lung cancer characteristics, adopting lung cancer screening can be strategy to detect lung cancer at early stages [6] and increase survival rates. According to the World Health Organization, screening and early diagnosis are two strategies for detecting cancer at early stages [7]. However, since lung cancer progresses fast and has unspecific symptoms, it is challenging to implement an early diagnosis strategy in health systems [4].

Low-dose computed tomography (LDCT) has been proven helpful for lung cancer screening. Two large clinical trials—National Lung Screening Trial (NLST) and Nederland's Leuven's Longkanker Screenings Onderzoek (NELSON)—have observed a decrease in 5 year mortality --in high-risk individuals using this screening strategy [8,9].

Despite these findings, LDCT is not adopted in many countries [10], mainly in those where there is a high incidence and prevalence of granulomatous diseases such as tuberculosis (TB) due to the high false positive rates [11]. On the other hand, pulmonary TB is considered an independent risk factor for lung cancer, especially in younger patients [12]. Nevertheless, evidence encourages LDCT screening for lung cancer, even in TB-endemic countries [13,14].

Some systematic reviews and meta-analyses have been performed to evaluate the LDCT screening for lung cancer and its association with mortality [15–18]. However, none have considered using LDCT screening for lung cancer in countries with a high incidence of granulomatous disease. Therefore, this synthesis is unique in obtaining more accurate and valid

estimates of LDCT screening for lung cancer and its effect on mortality, especially in TB-endemic countries.

## Scope of review

Our key questions (KQ) are:

KQ1. Does screening for lung cancer with LDCT change the incidence and distribution of lung cancer stages?

KQ2. Does the adoption of screening through LDCT decrease mortality from lung cancer in 18 years or older humans?

KQ2.1 If yes, in how many years?

KQ3. What is the rate of false-positive results found in these studies?

It is also intended to answer whether the adoption of LDCT has been analyzed in countries and/or territories with high incidence and/or prevalence of TB. If yes, was a decrease in mortality found in these countries and/or regions? Do they show changes in incidence and stage distribution? Is the rate of false positive results higher than in places with low incidence and/or prevalence of TB?

## Methods/Design

### Eligibility criteria

The tool used for eligibility criteria was the acronymous PICOS (Population, Intervention, Comparator/control, Outcomes, Study type) as a strategy to guide the research and create research questions and search strategy. No linguistic restriction was applied as part of the eligibility criteria. We analyzed studies published from 2010 to 2023.

### Search strategy

Structured terms were created based on information from PICOS that translated the search criteria into formulating a search strategy. We identified potentially relevant studies by searching multiple electronic databases and websites such as PubMed, Embase, Scopus, and BVS. Mesh terms and keywords related to screening, low-dose tomography computer and lung cancer were used. The search strategy was adapted for each database. The search terms were: Lung Neoplasms (ti (title), ab (abstract), kw (Keyword)) and/or lung cancer (ti, ab, kw) and Early Detection of Cancer (ti, ab, kw) and/or Mass Screening (ti, ab, kw) and/or screen*(ti, ab, kw).

### Ethics

This review does not require ethical approval as the review is based on the published data of the ethically approved primary studies. This study used secondary data available in the public domain, being exempt from ethical review by the Research Ethics Committee, according to the Brazilian National Ethics Committee (CONEP) and National Health Council (CNS) Resolutions 466/2012 and 510/2016.

### Study design

The protocol is registered in the PROSPERO database (International Prospective Register of Systematic Reviews) under CRD42022309581. Following the steps provided in the PRISMA-P guide (Preferred Reporting Items for Systematic Review and Meta-Analyses Protocol) [19].

A total of nine researchers participated in this review process. Eight of the nine were paired, considering experience in the literature review process and knowledge of the field of research. The one researcher who did not participate in the active review was assigned to evaluate the

divergences between the pair. All reviewers were trained using the extraction tool. The intermediate results were discussed among the team. In each step of the review process, the reasons for divergences between the two independent reviewers were evaluated and discussed in a feedback process.

The systematic review with meta-analysis was conducted following the recommendations of the Cochrane Collaboration Handbook of Systematic Reviews [20].

## Study selection

Two reviewers independently screened records for inclusion, applying eligibility criteria and selected studies for inclusion in the systematic review and researchers were blinded to each other's decisions. A third reviewer judged disagreements.

## Data extraction

We extracted the following information from the included studies: study design and methodology, participant demographics and baseline characteristics, numbers of events and measures of effect. Two individuals independently extracted data. A third reviewer judged disagreements. Missing data was recorded and analyzed in the quality report. Systematic Review Data Repository (SRDR+) was used to extract the metadata and record decisions.

The high prevalence/incidence of TB status was assessed using WHO's Global Tuberculosis Report 2022[21].

## Risk of bias (quality) assessment

Two authors independently assessed the included studies for risk of bias using the Cochrane "Risk of bias" assessment tool (Cochrane Handbook) to evaluate allocation (random sequence generation and allocation concealment), blinding of participants and personnel, blinding of outcome assessors; incomplete outcome data; and other potential sources of bias [22]. We resolved the disagreements by consensus.

Each domain was scored separately as low risk of bias, unclear risk of bias (insufficient information to make a judgment), or high risk of bias.

## Data synthesis and analysis

The interventions were described using the Template for Intervention Description and Replication (TIDIeR) checklist.

The studies were categorized into clinical trials–randomized control trials (RCT) and without control groups (CT)–and cohort observational studies. Studies that included countries/regions with a high incidence of TB was analyzed separately.

The results of screening studies may have been influenced by lead-time bias or overdiagnosis bias, giving rise to an apparent improvement in survival in the intervention group. Disease-specific mortality was, therefore, the primary outcome considered in the review.

We used incidence rate ratios (IRRs)method to assess if there is any evidence of the effect of LDCT screening on the outcomes. This was done separately for cohort studies and Clinical Trials (CT and RCT). We also used the method separately in studies with countries/regions with a high prevalence of TB.

For KQ1 and 2, forest plots were created to display the findings of each study by calculating incidence rate ratios (IRRs), using the number of events and person-years of follow-up for lung cancer incidence, lung cancer mortality, and all-cause mortality. For KQ3, we calculated

false positive frequency for each screening round. We used 95% Confidence Intervals (CI) in all analyses.

We performed a stratified analysis by the study type(cohort or clinical trials) and study population, identifying specifically countries or regions with a high incidence or prevalence of TB (due to the probability of false-positive results).

## Results

After evaluating 14,143 publications, 37 were included in this systematic review (Fig 1), pertaining to 22 different studies (10 RCT and 12 cohorts).

A breakdown of each study's characteristics is reported in Box 1.

Among the RCTs, seven trials were conducted in Europe (DANTE, DLCT, ITALUNG, LUSI, MILD, NELSON, UKLS), two in the USA (LSS and NLST) and one in China (CLUS). In the European and Chinese studies, the control group was the absence of screening (standard of care). In contrast, studies performed in the United States considered chest radiography (CXR) as its control. Only one RCT was conducted in a country with a high prevalence of TB (CLUS).

Sample size varied considerably among studies. In four RCTs, the mean participant age was within the 50–60 years old range (DLCST, LUSI, MILD, and NELSON), and at six, the mean age range was 60–70 (CLUS, DANTE, ITALUNG, LSS, NLST, UKLS). The population was predominantly male in all RCTs except CLUS (47% male). Most RCTs required smoking history as an inclusion criterion; the only exception was UKLS, which used a high score in a risk prediction model. The analyzed studies ranged from two to five screening rounds as follows: two (UKLS), three (CLUS, NLST), four (ITALUNG, NELSON), or five (DANTE, DLCST, LUSI, MILD) (Box 1).

The positive screen definition differed among the RCTs. CLUS and NLST considered positive non-calcified nodules with ≥4 mm. DANTE, DLCST, ITALUNG, and LUSI considered positive cases with ≥5 mm. LLS had different parameters for baseline (>3 mm) and follow-up (≥4 mm). MILD, NELSON, and UKLS used volumetric measurements of nodules (Box 1).

This systematic review included 12 cohort studies. Six of them were conducted in Asian countries (Hitachi cohort, Kaohsiung cohort, Nagano cohort, NLCSP cohort, Sungkyunkwan cohort, Taichung cohort), five in the USA (BLCS cohort, I-ELCAP cohort, Montefiore cohort, SEER-Medicare cohort, Veterans' Health Administration Cohort) and one in France (SOMME cohort). Nine cohort studies compared LDCT to no screening (standard of care) [48,50–53,55–57,60] and four to CXR [49,54,58,59].

Four cohort studies took place in countries with a high incidence or prevalence of TB [52,55,58,59]. Sample size varied greatly among them, and most studies' population mean age was in the 60–70 years range, except three studies with a population in the 50–60 bracket [54,58,59] and one [56] that didn't disclose that information. Regarding population distribution by gender, seven cohort studies had fewer male participants [48,49,53–57]. Two studies stood out for the female participation [54,57]. Three had more male participants [50,52,59], and two didn't disclose the information [51,58,60] (Box 1).

Montefiore [53] and Veterans' Health Administration [60] Cohorts required smoking history as an inclusion criterion. I-ELCAP [50] included individuals with smoking (median, 34 packyears), occupational (beryllium, radon, or uranium) or second-hand smoke exposure; BLCS [48], Hitachi [49], Kaohsiung [52], SEER-Medicare cohort [56], and SOMME [57] cohorts are retrospective cohorts with registries of patients diagnosed with LC, despite of smoking status; Nagano cohort [54] only included non-smokers; and NLCSP [55] and Sung-kyunkwan [58] cohort screened all participants but classified than in risk stratifications

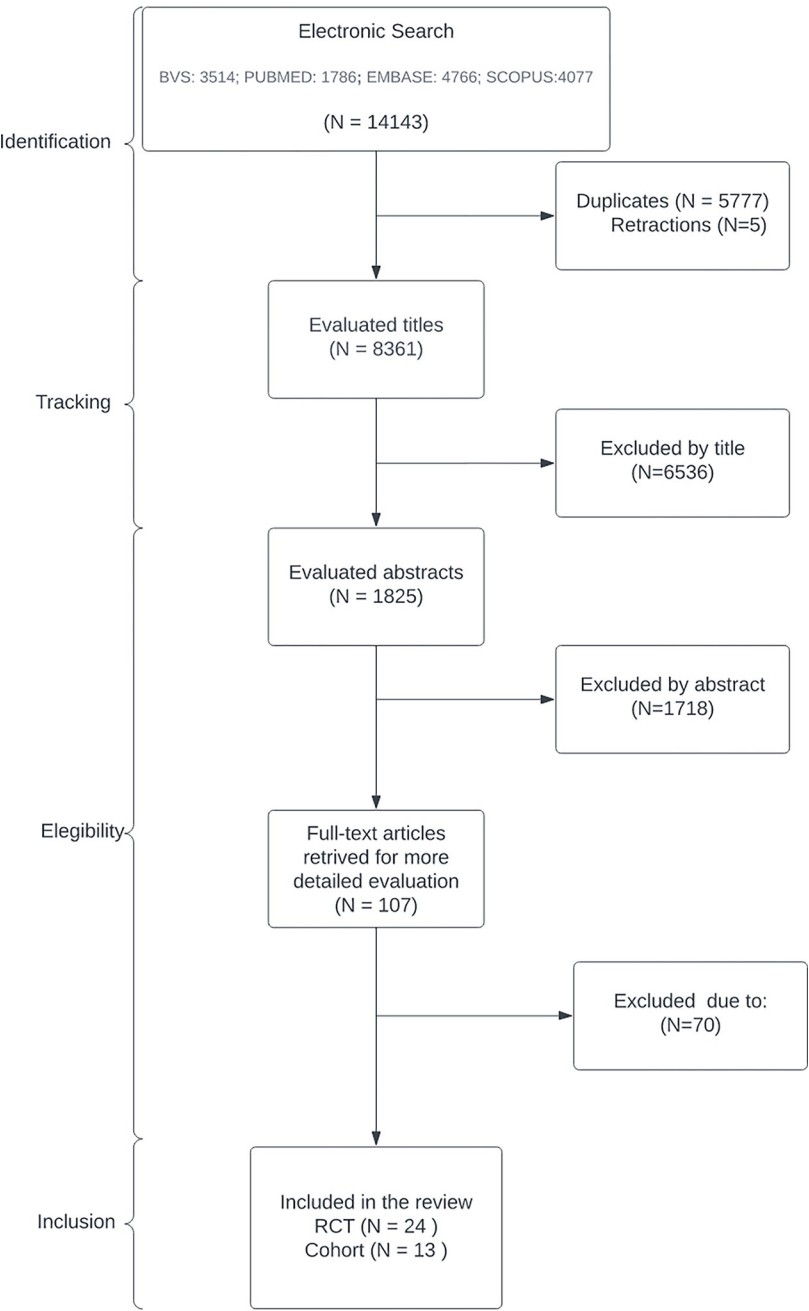

**Fig 1. PRISMA flow diagram.**

according to NLST (≥30 pack-years and ≥55 years of age) and European (≥20 pack-years and ≥50 years of age) risk stratification criteria. (Box 1)

Half the cohorts studied did not define a positive screen [48,51–54,56,60]. The six studies that did, it varied from ≥4 mm [59], ≥5 mm [57,58], and ≥8 mm [49]. I-ELCAP cohort [50] had different criteria for baseline (>6 mm) and annual screening (≥3 mm). (Box 1)

The overall risk of bias was low for 60% of the studies, while 40% had "some concerns". None of the ten evaluated RCTs had a high risk of bias (Fig 2).

## Box 1. Study characteristics, target population and screening characteristics

| Setting characteristics | Target population characteristics | | | | | Screening characteristics | | | | | | Follow-up (years) | Threshold of abnormal LN |
|---|---|---|---|---|---|---|---|---|---|---|---|---|---|
| Study, year | Study type | Comparison provided | Country | High TB incidence | Sample size (N) | Participation rate (%) | Mean age, Target age (years) | % Male | Smoking history required | Pack-years; years since quitting | Screening rounds (N) | Time between rounds (years) | | |
| CLUS[23], 2013–2018 | RCT | No screening | China | Yes | 6717 | 99.1 | 60 (45–70) | 47 | Yes | ≥20;<15 | 3 | 1 | 5 | ≥4 mm |
| DANTE[24,25], 2001–2013 | RCT | No screening | Italy | No | 2811 | 86.8 | 65 (60–74) | 100 | Yes | ≥20;<10 | 5 | 1 | 5 | ≥5 mm |
| DLCST[26,27], 2004–2016 | RCT | No screening | Denmark | No | 4104 | 99.3 | 58 (50–70) | 56 | Yes | ≥20; quit after 50 and <10 | 5 | 1 | 10 | ≥5 mm |
| ITALUNG[28,29], 2005–2014 | RCT | No screening | Italy | No | 3206 | 93.5 | 61 (55–69) | 65 | Yes | ≥20;<10 within the last 10 years | 4 | 1 | 9 | ≥5 mm |
| LSS[30,31], 2000–2007 | RCT | CXR | USA | No | 3318 | 94.5 | 64 (55–74) | 59 | Yes | ≥30;<10 | NI | 1 | 5 | Baseline: >3 mm Year 1: ≥4 mm |
| LUSI[32–34], 2007–2018 | RCT | No screening | Germany | No | 4052 | 99.7 | 55 (50–69) | 65 | Yes | ≥25 of 15 cigarettes/day or ≥30 of 10 cigarettes/day; <10 | 5 | 1 | 9 | ≥5 mm |
| MILD[25,35,36], 2005–2011 | RCT | No screening | Italy | No | 4099 | 99.9 | 57 (49+) | 66 | Yes | ≥20;<10 | 5(annual) 5 (biennial) | 1 or 0.5 | NI | >60 mm$^3$ |
| NELSON[37–39], 2000–2012 | RCT | No screening | Netherlands and Belgium | No | 15792 | 100 | 58 (50–74) | 84 | Yes | ≥25 of 15 cigarettes/day or ≥30 of 10 cigarettes/day; <10 | 4 | 1, 2, and 2.5 | 10 (and post-trial up-to 6 y) | >500 mm$^3$ |
| NLST[9,40–45], 2002–2009 | RCT | CXR | USA | No | 53454 | 98 | 61 (55–74) | 59 | Yes | ≥30; ≤15 | 3 | 1 | 7 (and post-trial up-to 12.3 y) | ≥4 mm |
| UKLS[46,47], 2011–2014 | RCT | No screening | UK | No | 4055 | 97.9 | 67 (50–74) | 75 | No‡ | - | 2 | 1 | 3 (and post-trial up-to 6 y) | >50mm$^3$ |
| BLCS cohort[48], 2013–2021 | COHORT | No screening | USA | No | 1216 | 100 | 67 | 44 | No¥ | - | NI | NI | 9 | NI |
| Hitachi cohort[49], 1998–2006 | COHORT | CXR | Japan | No | 33483 | 100 | 60 (50–74) | 49 | No¥ | - | NI | NI | 8 | ≥8 mm |
| I-ELCAP cohort[50,51], 1993–2009 | COHORT | No screening | USA | No | 48037 | 100 | 60 (40–85) | 58 | No‡ | - | 2 | 1 | 16 | Baseline: >6 mm Annual: ≥3 mm |
| Kaohsiung cohort[52], 2007–2017 | COHORT | No screening | Taiwan | Yes | 2883 | 100 | 66 (40–80) | 62 | No¥ | - | NI | NI | 10 | NI |
| Montefiore cohort[53], 2013–2018 | COHORT | No screening | USA | No | 175 | NI | 67 (55–80) | 48 | Yes | ≥30; ≤15 | 2 | 1 | 5 | NI |
| Nagano cohort[54], 2000–2008 | COHORT | CXR | Japan | No | 460 | 100 | 66 | 23 | No€ | - | 8 | 1 | 8 | NI |
| NLCSP cohort[55], 2013–2018 | COHORT | No screening | China | Yes | 1016740 | 100 | 56 (40–74) | 44 | No£ | - | 1 | - | 3.6 | NI |
| SEER-Medicare cohort[56], 2015–2019 | COHORT | No screening | USA | No | 414358 | 100 | (65–77) | 45 | No¥ | - | - | - | 3.6 | NI |
| SOMME cohort[57], 2016–2017 | COHORT | No screening | France | No | 664 | 72.6 | 63 (55–74) | 28 | No¥ | ≥30; ≤15 just for the screened group | 2 | 1 | 2 | ≥5 mm |
| Sungkyunkwan cohort[58], 2006–2008 | COHORT | CXR | South Korea | Yes | 12427 | 97.9 | 52 (16–90) | NI | No£ | - | 3 | 1 | 3 | ≥5 mm |
| Taichung cohort[59], 2012–2013 | COHORT | CXR | Taiwan | Yes | 3339 | NI | 48 (18+) | 52.3 | No | - | 1 | - | 1 | ≥4 mm |
| Veterans' Health Administration Cohort[60], 2015–2017 | COHORT | No screening | USA | No | 4664 | 100 | 68 (55–80) | NI | Yes | No criteria | NI | NI | NI | NI |

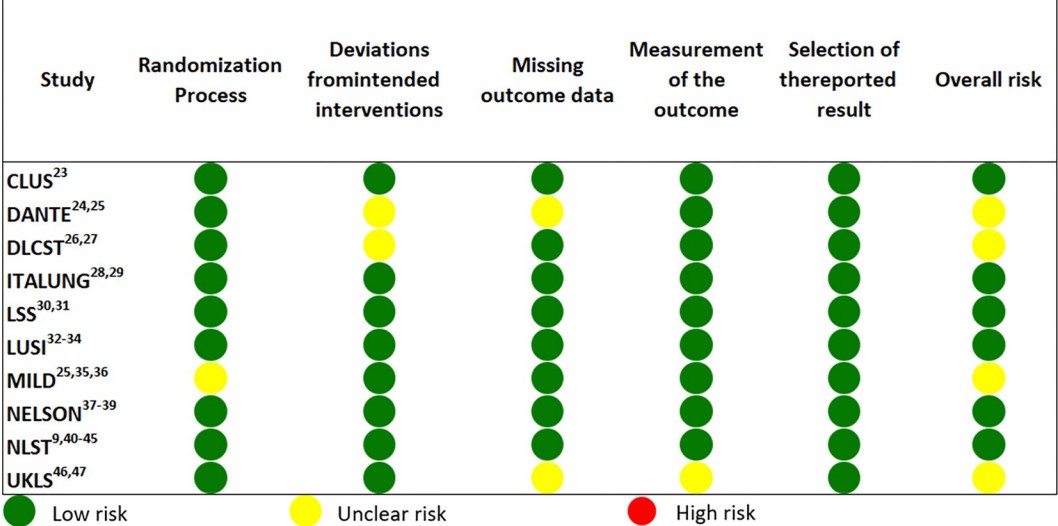

**Fig 2. Risk of bias assessment of RCTs included in the systematic review.** *Risk of boas assessed with version 2 of the Cochrane risk-of-bias tool for randomized trials (RoB 2).

Among the twelve cohort studies included in this systematic review four had a low overall risk of bias, seven had "some concerns", and one study was evaluated as having a high risk of bias (Fig 3).

Table 1 and Fig 4 show the results of KQ1: *"Does screening for lung cancer with LDCT change the incidence and distribution of lung cancer stages?"* Ten RCTs (described in 25 articles) and nine cohort studies were included. Some studies divided their results by risk-based categories [23,55,58] or screening periodicity [36].

Cumulative incidence for LC was higher in the LDCT group in all RCTs except MILD (both annual and biennial). All studies noticed an increase in early-stage (I-II) diagnoses in the LDCT group. Similarly, almost all trials found that late-stage (III-IV) LC incidence is significantly lower in the LCDT arm when compared to control. The only exception is DLCST, whose IRR favored control. It's interesting to notice that the trial conducted in a country with high TB incidence (CLUS–China) had similar results to those in Europe and the USA. (Table 1 and Fig 4).

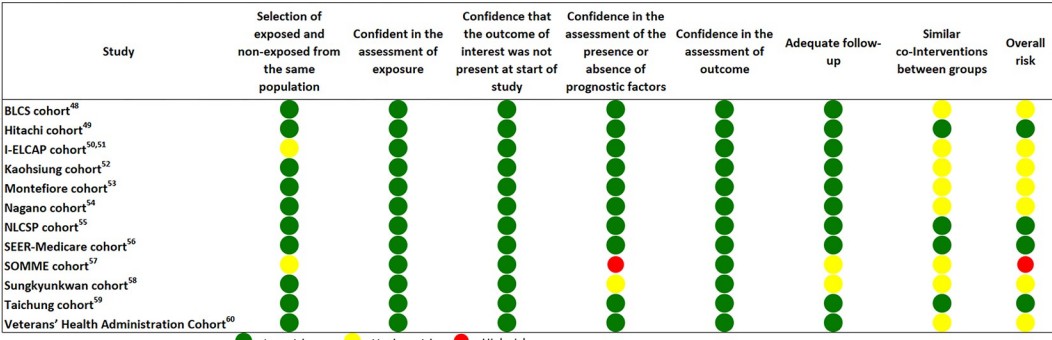

**Fig 3. Risk of bias assessment of cohorts included in the systematic review.** *Risk of boas assessed with Cochrane Risk Of Bias in Non-randomized Studies–of interventions (ROBINS-I tool).

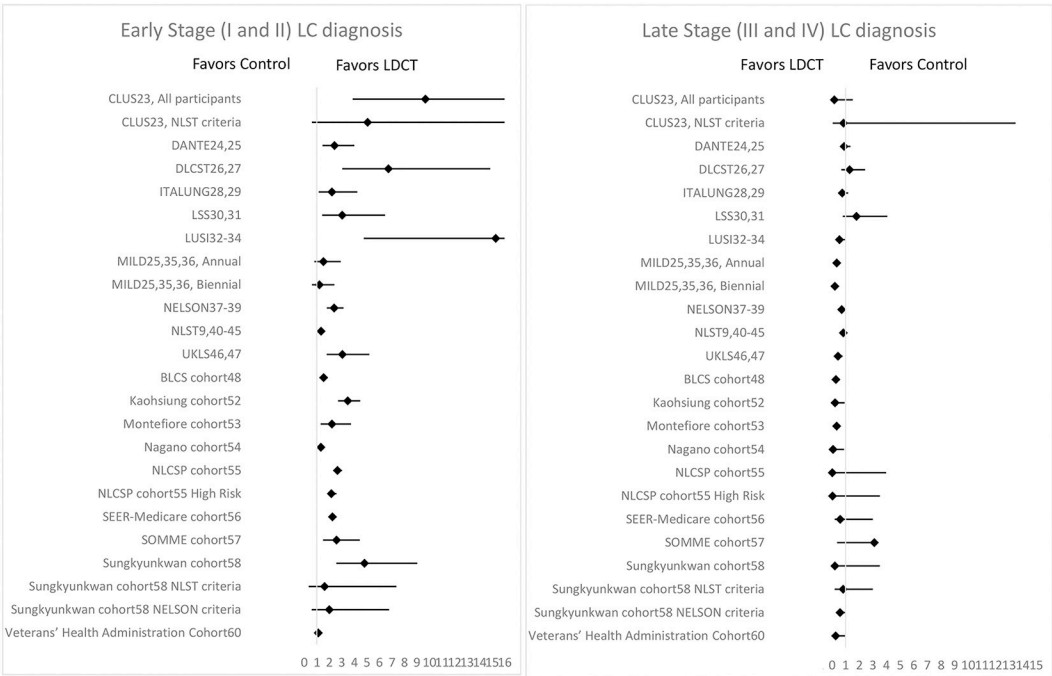

**Fig 4. Results of incidence rate ratio (IRR) for incidence of Earty- (l-II) and Late- (Ill-IV) Satge Lung Cancer (CI 95˚).**

Cohort studies had similar results to the RCTs. The majority showed more LC diagnosed in the LDCT arm [48,55–58], 3 studies showed no difference in IRR among groups [53,54,60], and in 1 more LC was found in the control group [52]. All studies found that early-stage diagnosis was higher and late-stage diagnosis was lower in the LDCT arm compared to the control, except in Sungkyunkwan cohort [58], in which higher late-stage diagnosis was also found in the LDCT arm. Again, studies conducted in regions with a high TB incidence had results on par with their counterparts (Table 1 and Fig 4).

Table 2 and Fig 5 detail the results of KQ2: *"Does the adoption of screening through LDCT decrease mortality from lung cancer in 18 years or older humans?"* Nine RCTs (described in 24 articles) and 3 cohort studies were included on LC-specific mortality, and 9 RCTs and 2 cohort studies contributed information on all-cause mortality.

Five RCTs reported reduced LC mortality on the LDCT arm compared to the control (ITA-LUNG, LUSI, NELSON, NLST, UKLS). DANTE, DLCST, and MILD showed little difference among groups. LSS results were unique and reported increased IRR for the LDCT group. On all causes of mortality outcome, DANTE and ITALUNG encountered reduced IRR on the LDCT arm. DLCST, LUSI, NELSON, NLST, and UKLS reported few differences between LDCT and control arms. LSS and MILD showed increased IRR in the LDCT group (Table 2 and Fig 5).

All cohorts pointed out lower IRR in the LDCT group, both for LC and all causes of mortality. (Table 2 and Fig 5) The only study conducted in a high-incidence TB country was NLCSP cohort [55] (China). The results on reduced mortality (LC and all causes) are akin to the other cohorts and RCTs.

Table 3 details the results for KQ3: *"What is the rate of false-positive results found in these studies?"* Ten RCTs (described in 25 articles) and 1 cohort study reported enough information to determine the rate of false positives, defined as any result leading to additional evaluation

**Table 1. RCT and Cohort Results of Incidence of Early- (I-II) and Late- (III-IV) Stage Lung Cancer.**

| Study | High TB | Mean age, years | Mean pack-years | Screening-times | N participants | | Early-stage lung cancer (I-II) | | | Late-stage lung cancer (III-IV) | | | All diagnosis | | |
|---|---|---|---|---|---|---|---|---|---|---|---|---|---|---|---|
| | | | | | LDCT | Control | LDCT | Control | IRR (95% CI) | LDCT | Control | IRR (95% CI) | LDCT | Control | IRR (95% CI) |
| Randomized Clinical Trials (RCT) | | | | | | | | | | | | | | | |
| CLUS [23], All participants | Yes | 60 | - | 3 | 3512 | 3145 | 54 | 5 | 9.67 (3.87 to 24.18) | 1 | 5 | 0.18 (0.02 to 1.53) | 55 | 10 | 4.93 (2.51 to 9.66) |
| CLUS [23], NLST criteria | Yes | 60 | - | 3 | 256 | 216 | 6 | 1 | 5.06 (0.61 to 42.05) | 1 | 1 | 0.84 (0.05 to 13.5) | 7 | 2 | 2.95 (0.61 to 14.2) |
| DANTE [24,25] | No | 65 | 47 | 5 | 1264 | 1186 | 54 | 21 | 2.41 (1.46 to 3.99) | 43 | 45 | 0.9 (0.59 to 1.36) | 97 | 66 | 1.38 (1.01 to 1.89) |
| DLCST [26,27] | No | 58 | 36 | 5 | 2052 | 2052 | 47 | 7 | 6.71 (3.03 to 14.85) | 22 | 17 | 1.29 (0.69 to 2.44) | 69 | 24 | 2.88 (1.81 to 4.57) |
| ITALUNG [28,29] | No | 61 | 39 | 4 | 1613 | 1593 | 29 | 13 | 2.2 (1.15 to 4.24) | 33 | 43 | 0.76 (0.48 to 1.19) | 62 | 56 | 1.09 (0.76 to 1.57) |
| LSS [30,31] | No | - | 54 | - | 1629 | 1648 | 27 | 9 | 3.03 (1.43 to 6.45) | 16 | 9 | 1.8 (0.79 to 4.07) | 43 | 18 | 2.42 (1.39 to 4.19) |
| LUSI [32–34] | No | 55 | - | 5 | 2029 | 2023 | 46 | 3 | 15.29 (4.75 to 49.16) | 16 | 29 | 0.55 (0.3 to 1.01) | 62 | 32 | 1.93 (1.26 to 2.96) |
| MILD [25,35,36], Annual | No | 57 | 38 | 7 | 1190 | 1723 | 19 | 18 | 1.53 (0.8 to 2.91) | 10 | 42 | 0.34 (0.17 to 0.69) | 29 | 60 | 0.7 (0.45 to 1.09) |
| MILD [25,35,36], Biennial | No | 57 | 38 | 7 | 1186 | 1723 | 15 | 18 | 1.21 (0.61 to 2.4) | 6 | 42 | 0.21 (0.09 to 0.49) | 21 | 60 | 0.51 (0.31 to 0.84) |
| NELSON [37–39] | No | 58 | 38 | 4 | 6583 | 6612 | 168 | 71 | 2.38 (1.8 to 3.14) | 153 | 216 | 0.71 (0.58 to 0.88) | 321 | 287 | 1.12 (0.96 to 1.32) |
| NLST [9,40–45] | No | 61 | 56 | 3 | 26722 | 26732 | 818 | 615 | 1.33 (1.2 to 1.48) | 766 | 918 | 0.83 (0.76 to 0.92) | 1584 | 1533 | 1.03 (0.96 to 1.11) |
| UKLS [46,47] | No | 68 | - | 2 | 1994 | 2027 | 54 | 18 | 3.05 (1.79 to 5.2) | 16 | 37 | 0.44 (0.24 to 0.79) | 70 | 55 | 1.29 (0.91 to 1.84) |
| Cohort Studies | | | | | | | | | | | | | | | |

(*Continued*)

**Table 1.** (Continued)

| Study | High TB | Mean age, years | Mean pack-years | Screening-times | N participants | | Early-stage lung cancer (I-II) | | | Late-stage lung cancer (III-IV) | | | All diagnosis | | |
|---|---|---|---|---|---|---|---|---|---|---|---|---|---|---|---|
| | | | | | LDCT | Control | LDCT | Control | IRR (95% CI) | LDCT | Control | IRR (95% CI) | LDCT | Control | IRR (95% CI) |
| BLCS cohort [48] | No | 67 | - | - | 114 | 1102 | 99 | 620 | 1.54 (1.25 to 1.91) | 12 | 427 | 0.27 (0.15 to 0.48) | 111 | 1047 | 1.02 (0.84 to 1.25) |
| Kaohsiung cohort [52] | Yes | 69 | - | - | 93 | 2790 | 67 | 579 | 3.47 (2.7 to 4.47) | 12 | 2207 | 0.16 (0.09 to 0.29) | 79 | 2786 | 0.85 (0.68 to 1.06) |
| Montefiore cohort [53] | No | 67 | 50 | 2 | 33 | 142 | 21 | 41 | 2.2 (1.3 to 3.73) | 12 | 101 | 0.51 (0.28 to 0.93) | 33 | 142 | 1 (0.68 to 1.46) |
| Nagano cohort [54] | No | 66 | - | 8 | 218 | 160 | 210 | 116 | 1.33 (1.06 to 1.67) | 8 | 44 | 0.13 (0.06 to 0.28) | 218 | 160 | 1 (0.82 to 1.23) |
| NLCSP cohort [55] | Yes | 56 | - | 1 | 79581 | 937159 | 271 | 1206 | 2.65 (2.32 to 3.02) | 118 | 836 | 1.66 (1.37 to 2.02) | 389 | 2042 | 2.24 (2.01 to 2.5) |
| NLCSP cohort [55] High Risk | Yes | 56 | - | 1 | 79581 | 143721 | 271 | 227 | 2.16 (1.81 to 2.57) | 118 | 221 | 0.96 (0.77 to 1.21) | 389 | 448 | 1.57 (1.37 to 1.8) |
| SEER-Medicare cohort [56] | No | - | - | - | 7336 | 407022 | 473 | 11692 | 2.24 (2.05 to 2.46) | 275 | 21539 | 0.71 (0.63 to 0.8) | 1150 | 47741 | 1.34 (1.26 to 1.42) |
| SOMME cohort [57] | No | 65 | 42 | 2 | 18 | 626 | 14 | 189 | 2.58 (1.5 to 4.43) | 4 | 420 | 0.33 (0.12 to 0.89) | 18 | 609 | 1.03 (0.64 to 1.64) |
| Sungkyunkwan cohort [58] | Yes | 52 | 20 | 3 | 5771 | 6656 | 50 | 12 | 4.81 (2.56 to 9.02) | 9 | 7 | 1.48 (0.55 to 3.98) | 59 | 19 | 3.58 (2.14 to 6.01) |
| Sungkyunkwan cohort [58] NLST criteria | Yes | - | - | 3 | 903 | 291 | 10 | 2 | 1.61 (0.35 to 7.35) | 4 | 2 | 0.64 (0.12 to 3.52) | 14 | 4 | 1.13 (0.37 to 3.43) |
| Sungkyunkwan cohort [58] NELSON criteria | Yes | - | - | 3 | 1928 | 641 | 18 | 3 | 1.99 (0.59 to 6.77) | 7 | 3 | 0.78 (0.2 to 3) | 25 | 6 | 1.39 (0.57 to 3.38) |
| Veterans' Health Administration Cohort [60] | No | 68 | - | - | 118 | 4546 | 101 | 3476 | 1.12 (0.92 to 1.36) | 17 | 1070 | 0.61 (0.38 to 0.99) | 118 | 4546 | 1 (0.83 to 1.2) |

(eg, repeat LDCT scan before the next annual screening, biopsy) that did not result in a cancer diagnosis. The results are presented by screening rounds, and MILD trial is divided by screening periodicity [36].

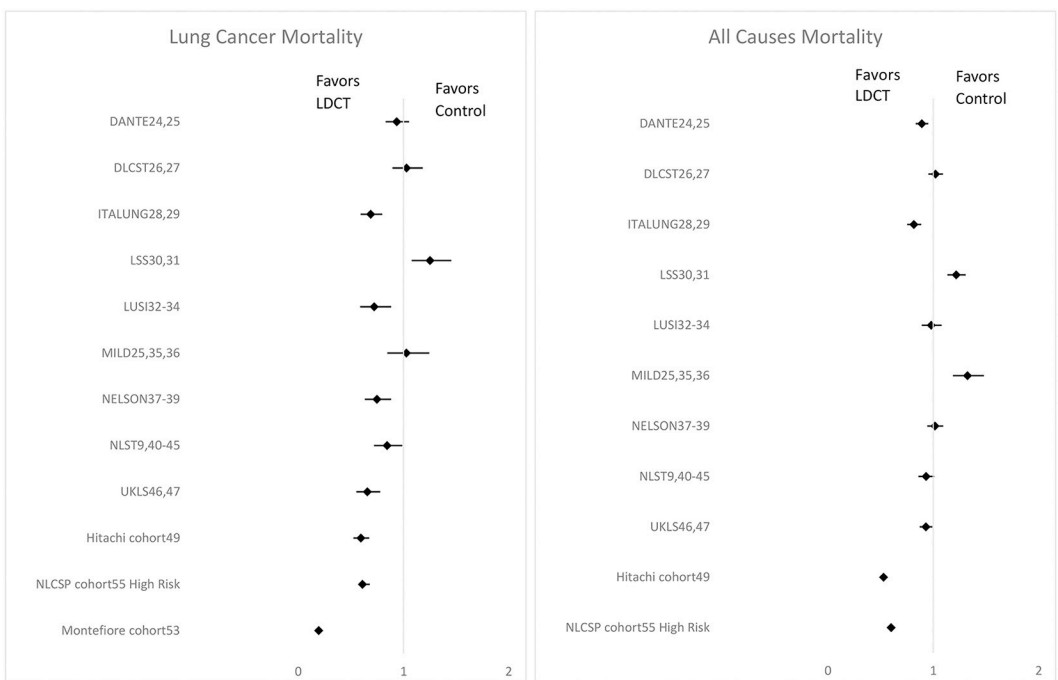

**Fig 5. Results of incidence rate ratio (IRR) for lung cancer and all-cause mortality (CI 95%).**

False-positive rates varied widely across trials, most likely because of differences in definitions of positive results, such as cutoffs for nodule size and use of volume-doubling time. The range of false-positive overall was 7.9% to 29.0% for baseline. Despite that, most studies stayed within the 20 to 30% range (CLUS, ITALUNG, LUSI, NELSON, NLST, UKLS). False-positive rates generally declined with each screening round (DLCST, ITALUNG, LUSI, MILD, NELSON, NLST). It's interesting to notice that the Chinese trial had false-positive rates well within the range of NLST and NELSON trials. The Taichung cohort, the only cohort study that reported the false positive rate, found 37.3% (35.6 to 38.9).

## Discussion

RCT and cohort studies had compatible results. The majority noticed an increase in early-stage (I-II) diagnoses in the LDCT group. Similarly, almost all trials found that late-stage (III-IV) LC incidence is significantly lower in the LCDT arm when compared to control. Most analyses reported reduced LC mortality in the LDCT arm compared to the control. Although false-positive rates varied among papers, most studies stayed within the 20 to 30% range. False-positive rates generally declined significantly with each screening round. All these results are in accordance with other systematic reviews done on this subject (15–18).

Only five studies (1 RCT and four cohorts) that meet this review's standards took place in countries with a high prevalence of TB. The RCT and one of the cohorts were evaluated as a low risk of bias for all evaluated components and the other three cohort studies had "some concerns". Another Clinical Trial is currently happening in South Korea [61], and will render more reliable information on the benefits of LC screening with LDCT in TB-endemic countries. Initial results of this study showed that, although the specificity of Lung-RADS was higher in participants without tuberculosis sequelae than in those with tuberculosis sequelae (85% vs 80%, respectively; P, 0.001), the difference does not impede implementing LDCT lung

**Table 2. RCT and cohort results of lung cancer mortality and all causes mortality.**

| Study | High TB | Mean age, years | Mean pack-years | Screening-times | N participants | | Lung Cancer Mortality (per 100,000 person-years) | | | | | All causes Mortality per 100,000 person-years | | | | |
|---|---|---|---|---|---|---|---|---|---|---|---|---|---|---|---|---|
| | | | | | LDCT | Control | LDCT (N) | Control (N) | LDCT | Control | IRR (95% CI) | LDCT (N) | Control (N) | LDCT | Control | IRR (95% CI) |
| **Randomized Clinical Trials (RCT)** | | | | | | | | | | | | | | | | |
| DANTE [24,25] | No | 65 | 47 | 5 | 1264 | 1186 | 59 | 55 | 543 | 544 | 0.94 (0.83 to 1.05) | 180 | 176 | 1655 | 1742 | 0.89 (0.83 to 0.95) |
| DLCST [26,27] | No | 58 | 36 | 5 | 2052 | 2052 | 39 | 38 | 399 | 388 | 1.03 (0.89 to 1.18) | 166 | 163 | 1699 | 1664 | 1.02 (0.95 to 1.09) |
| ITALUNG [28,29] | No | 61 | 39 | 4 | 1613 | 1593 | 43 | 60 | 293 | 421 | 0.69 (0.59 to 0.8) | 154 | 181 | 1051 | 1270 | 0.82 (0.75 to 0.89) |
| LSS [30,31] | No | - | 54 | - | 1629 | 1648 | 32 | 26 | 384 | 310 | 1.25 (1.08 to 1.45) | 139 | 116 | 1667 | 1384 | 1.22 (1.13 to 1.31) |
| LUSI [32–34] | No | 55 | - | 5 | 2029 | 2023 | 29 | 40 | 161 | 222 | 0.72 (0.59 to 0.88) | 148 | 150 | 820 | 834 | 0.98 (0.89 to 1.08) |
| MILD [25,35,36] | No | 57 | 38 | 7 | 1190 | 1723 | 40 | 40 | 176 | 248 | 1.03 (0.85 to 1.25) | 137 | 106 | 603 | 658 | 1.33 (1.19 to 1.48) |
| NELSON [12,13] | No | 58 | 38 | 4 | 6583 | 6612 | 181 | 242 | 241 | 324 | 0.75 (0.63 to 0.88) | 868 | 860 | 1393 | 1376 | 1.02 (0.94 to 1.1) |
| NLST [9,40–45] | No | 61 | 56 | 3 | 26722 | 26732 | 469 | 552 | 280 | 332 | 0.84 (0.72 to 0.99) | 1912 | 2039 | 1141 | 1225 | 0.93 (0.86 to 1.01) |
| UKLS [46,47] | No | 68 | - | 2 | 1994 | 2027 | 30 | 46 | 213 | 330 | 0.66 (0.55 to 0.78) | 246 | 266 | 1748 | 1911 | 0.93 (0.87 to 0.99) |
| **Cohort Studies** | | | | | | | | | | | | | | | | |
| Hitachi cohort [49] | No | 62 | - | - | 17935 | 15548 | 72 | 80 | 408 | 595 | 0.59 (0.52 to 0.67) | 885 | 1188 | 5012 | 8829 | 0.53 (0.51 to 0.55) |
| NLCSP cohort [55] High Risk | Yes | 56 | - | 1 | 79581 | 143721 | 76 | 218 | 27 | 44 | 0.61 (0.58 to 0.65) | 176 | 515 | 62 | 103 | 0.60 (0.59 to 0.62) |
| Montefiore cohort [53] | No | 67 | 50 | 2 | 33 | 142 | 8 | 76 | 485 | 10704 | 0.19 (0.18 to 0.21) | - | - | - | - | - |

**Table 3. RCT and Cohort Results for false positive results, per round of screening.**

| | | | | | RCT Studies | | | | | | | Cohort Studies | |
|---|---|---|---|---|---|---|---|---|---|---|---|---|---|
| Study | | CLUS[23] | DANTE[24,25] | DLCST[26,27] | ITALUNG[28,29] | LSS[30,31] | LUSI[32-34] | MILD[25,35,36] Annual | MILD[25,35,36] Biennial | NELSON[37-39] | NLST[9,40-45] | UKLS[46,47] | Taichung cohort[59] |
| Mean age, years | | 60 | 65 | 58 | 61 | - | 55 | 57 | 57 | 58 | 61 | 68 | 48 |
| Mean pack-years | | - | 47 | 36 | 39 | 54 | - | 38 | 38 | 38 | 56 | - | - |
| Screening-times | | 3 | 5 | 5 | 4 | - | 5 | 7 | 7 | 4 | 3 | 2 | 1 |
| High TB | | Yes | No | No | No | No | No | No | No | No | No | No | Yes |
| Study threshold of na abnormal non-calcified lung nodule (screening test positive) | | ≥4 mm | ≥5 mm | ≥5 mm | ≥4 mm | Baseline: >3 mm Year 1: ≥4 mm | ≥5 mm | >60 mm³ | >60 mm³ | >500 mm³ | ≥4 mm | >50mm³ | ≥4 mm |
| **Baseline** | Number screened | 3512 | 1264 | 2047 | 1406 | 1556 | 2028 | 1152 | 1151 | 6583 | 26309 | 1994 | 3339 |
| | First image abnormal findings (LDCT) | 804 | 169 | 179 | 426 | 295 | 452 | 160 | 158 | 1570 | 7191 | 564 | 1279 |
| | True positive diagnosis (LDCT) | 55 | 29 | 17 | 18 | 30 | 24 | 11 | 6 | 70 | 270 | 70 | 34 |
| | False positive (95% IC) | 21.3 (20 to 22.7) | 11.1 (10.1 to 12.8) | 7.9 (6.7 to 9.1) | 29.0 (26.6 to 31.4) | 25.6 (23.3 to 27.9) | 21.1 (19.3 to 22.9) | 12.9 (11.0 to 14.9) | 13.2 (11.3 to 15.2) | 22.8 (21.8 to 23.8) | 26.3 (25.8 to 26.8) | 24.8 (22.9 to 26.7) | 37.3 (35.6 to 38.9) |
| **Round 1** | Number screened | | 1260 | 1976 | 1356 | 1374 | 1892 | 1111 | 147 | 6583 | 24715 | | |
| | First image abnormal findings (LDCT) | | 187 | 45 | 234 | 360 | 90 | 31 | 5 | 576 | 6901 | | |
| | True positive diagnosis (LDCT) | | 37 | 11 | 2 | 8 | 11 | 5 | 2 | 55 | 168 | | |
| | False positive frequency (95% IC) | | 11.9 (10.1 to 13.7) | 1.7 (1.1 to 2.3) | 17.1 (15.1 to 19.1) | 17.0 (15.2 to 18.9) | 4.2 (3.3 to 5.1) | 2.3 (1.5 to 3.2) | 2.0 (-0.2 to 4.3) | 7.9 (7.3 to 8.6) | 27.2 (26.7 to 27.8) | | |
| **Round 2** | Number screened | | | 1944 | 1308 | | 1849 | 1086 | 1086 | 6583 | 24102 | | |
| | First image abnormal findings (LDCT) | | | 52 | 211 | | 79 | 48 | 51 | 250 | 4054 | | |
| | True positive diagnosis (LDCT) | | | 13 | 9 | | 11 | 5 | 5 | 75 | 211 | | |
| | False positive frequency (95% IC) | | | 2.0 (1.4 to 2.6) | 15.4 (13.5 to 17.4) | | 3.7 (2.8 to 4.5) | 4.0 (2.8 to 5.1) | 4.2 (3.0 to 5.4) | 2.7 (2.3 to 3.0) | 15.9 (15.5 to 16.4) | | |

*(Continued)*

**Table 3.** (Continued)

| Study | | CLUS[23] | DANTE[24,25] | DLCST[26,27] | ITALUNG[28,29] | LSS[30,31] | LUSI[32-34] | MILD[25,35,36] Annual | MILD[25,35,36] Biennial | NELSON[37-39] | NLST[9,40-45] | UKLS[46,47] | Taichung cohort[59] |
|---|---|---|---|---|---|---|---|---|---|---|---|---|---|
| | | | | | | | | | | RCT Studies | | | Cohort Studies |
| Round 3 | Number screened | | | 1982 | 1263 | | 1826 | 1045 | 163 | | | | |
| | First image abnormal findings (LDCT) | | | 44 | 173 | | 113 | 25 | 13 | | | | |
| | True positive diagnosis (LDCT) | | | 12 | 6 | | 9 | 4 | 4 | | | | |
| | False positive frequency (95% IC) | | | 1.6 (1.1 to 2.2) | 13.2 (11.4 to 15.1) | | 5.7 (4.6 to 6.8) | 2.0 (1.2 to 2.9) | 5.5 (2.0 to 9.0) | | | | |
| Round 4 | Number screened | | | 1851 | | | 1565 | 1004 | 983 | | | | |
| | First image abnormal findings (LDCT) | | | 51 | | | 99 | 18 | 31 | | | | |
| | True positive diagnosis (LDCT) | | | 16 | | | 7 | 3 | 5 | | | | |
| | False positive frequency (95% IC) | | | 1.9 (1.3 to 2.5) | | | 5.9 (4.7 to 7.0) | 1.5 (0.7 to 2.2) | 2.6 (1.6 to 3.6) | | | | |
| Round 5 | Number screened | | | | | | | 795 | 157 | | | | |
| | First image abnormal findings (LDCT) | | | | | | | 5 | 12 | | | | |
| | True positive diagnosis (LDCT) | | | | | | | 2 | 2 | | | | |
| | False positive frequency (95% IC) | | | | | | | 0.4 (0 to 0.8) | 6.4 (2.5 to 10.2) | | | | |

(Continued)

**Table 3.** (Continued)

| Study | | | CLUS[23] | DANTE[24,25] | DLCST[26,27] | ITALUNG[28,29] | LSS[30,31] | LUSI[32-34] | MILD[25,35,36] Annual | MILD[25,35,36] Biennial | NELSON[37-39] | NLST[9,40-45] | UKLS[46,47] | Taichung cohort[59] |
|---|---|---|---|---|---|---|---|---|---|---|---|---|---|---|
| | | | | | | | | | **RCT Studies** | | | | | **Cohort Studies** |
| Round 6 | Number screened | | | | | | | | 428 | 751 | | | | |
| | First image abnormal findings (LDCT) | | | | | | | | 15 | 34 | | | | |
| | True positive diagnosis (LDCT) | | | | | | | | 4 | 1 | | | | |
| | False positive frequency (95% IC) | | | | | | | | 2.6 (1.1 to 4.1) | 4.4 (2.9 to 5.9) | | | | |

cancer screening in territories with high TB incidence/prevalence. This study is the first meta-analysis of LDCT for lung cancer screening that focuses on the procedure's benefits in countries with a high incidence/prevalence of TB. This investigation is necessary because, among the issues commonly raised when dealing with the implementation of lung cancer screening in these territories, the most relevant is a possible increase in the probability of false-positive results due to the high prevalence of tuberculosis, generating radiological images that create challenges in the differential diagnosis, which would mean that the procedure would not have an impact on the incidence and mortality in these locations.

As we have seen, the results indicate that lung cancer screening with LDCT generated results similar to those in European countries and the USA. In addition to the fact that there are not enough studies carried out in such environments for us to have conclusive results, all of them were carried out in Asian countries. Although this is the region with the highest incidence (234 per 100,000 inhabitants), the disease also has a considerable burden in other areas, such as the African continent (212 per 100,000) and the Eastern Mediterranean Region (112 per 100,000), as well as in countries like Brazil (40 per 100,000) [21]. Due to the ethnic diversity of the regions, a factor known to impact the risk of developing lung cancer [62], it would be essential to have studies with control parameters carried out in other regions of the world.

Recent data from Asia-Pacific countries demonstrates an alarming disparity in lung cancer patterns compared to Western countries, including a strikingly high proportion of lung cancer detected among never-smokers [63]. Although local data trends are still under analysis, there may be evidence of a fundamental difference in genetic predisposition among the Asian population that could impair the comparability of these results.

It is interesting to note that there are studies carried out in other countries with a high incidence of TB, such as India [64], Russia [65], and Brazil [66]. These studies were not included in the present review because they did not have a control arm, but their results in incidence per stage and false positives align with what was found in the included articles [11,66].

Furthermore, new technologies can alleviate the adverse effects of false positives in LC screening with LDCT. Liquid biopsy [67] and using Artificial Intelligence [68] in imaging analysis are promising. Nonetheless, implementing these new technologies still needs more cost-effectiveness studies in different settings [69,70].

In this sense, unlike other reviews [15,17,18,71–75], the option of including cohort studies in this analysis is justified since RCTs are more expensive and not so easily financed in low- and middle-income countries (LMIC) where the incidence of TB is higher.

One notable strength of this study is the inclusion of cohorts from LMICs, which significantly enhances the finding's external validity. LMICs often comprise a more extensive and diverse population base, making them an essential demographic to consider in research. Moreover, the fact that the results obtained from these cohorts align closely with those of RCTs is particularly compelling. Such consistency between different study designs bolsters the robustness of the conclusions and underscores the real-world applicability of the interventions or outcomes under investigation. It lends greater credence to the study's findings and emphasizes its capacity to inform policies and practices with global relevance.

It is relevant to mention that a re-analysis of the original data from the retained studies was not performed and can be considered a study limitation. Nonetheless, we aggregate and synthesize the results reported in the studies included in the analysis. The Meta-analysis provides a quantitative estimate or summary of the findings from multiple studies on a particular topic, allowing researchers to draw more robust conclusions than those based on individual studies alone.

We were able to obtain all full text included in the review. One challenge presented in this review was that many of the studies were excluded because they were implementation studies, and many were performed in countries with a high incidence/prevalence of TB.

Insights from this literature review may inform future guidelines on delivering LDCT screening in low- and middle-income countries (LMICs) with high TB burden. Our results emphasize the importance of implementing structured screening programs. Future studies should approach a scoping literature review focusing on Lung Cancer Screening Implementation in high-incidence and prevalence TB settings. Furthermore, another aspect noted was that the type of health care (universal, insurance base, among others) could impact how the technology is adopted in a specific country and who could benefit from its implementation.

## Supporting information

**S1 Checklist. PRISMA 2020 checklist.**
(DOCX)

## Author Contributions

**Conceptualization:** Debora Castanheira Pires, Luisa Arueira Chaves, Mario Jorge Sobreira da Silva, Mônica Rodrigues Campos, Isabel Cristina Martins Emmerick.

**Data curation:** Debora Castanheira Pires, Carlos Henrique Dantas Cardoso, Lara Vinhal Faria, Silvio Rodrigues Campos, Mario Jorge Sobreira da Silva, Tayna Sequeira Valerio.

**Formal analysis:** Debora Castanheira Pires, Luisa Arueira Chaves, Carlos Henrique Dantas Cardoso, Lara Vinhal Faria, Silvio Rodrigues Campos, Mario Jorge Sobreira da Silva, Tayna Sequeira Valerio, Mônica Rodrigues Campos, Isabel Cristina Martins Emmerick.

**Funding acquisition:** Luisa Arueira Chaves, Mônica Rodrigues Campos.

**Investigation:** Isabel Cristina Martins Emmerick.

**Methodology:** Debora Castanheira Pires, Luisa Arueira Chaves, Carlos Henrique Dantas Cardoso, Lara Vinhal Faria, Silvio Rodrigues Campos, Mario Jorge Sobreira da Silva, Tayna Sequeira Valerio, Mônica Rodrigues Campos, Isabel Cristina Martins Emmerick.

**Project administration:** Debora Castanheira Pires, Luisa Arueira Chaves.

**Writing – original draft:** Debora Castanheira Pires, Luisa Arueira Chaves, Mario Jorge Sobreira da Silva, Mônica Rodrigues Campos, Isabel Cristina Martins Emmerick.

**Writing – review & editing:** Debora Castanheira Pires, Luisa Arueira Chaves, Carlos Henrique Dantas Cardoso, Lara Vinhal Faria, Silvio Rodrigues Campos, Mario Jorge Sobreira da Silva, Tayna Sequeira Valerio, Mônica Rodrigues Campos, Isabel Cristina Martins Emmerick.

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
