## [Decision Letter · Decision Letter 0]

15 Feb 2024

PONE-D-23-38324Effects of Low Dose Computed Tomography (LDCT) on lung cancer screening on incidence and mortality in regions with high tuberculosis prevalence: a systematic reviewPLOS ONE

Dear Dr. Emmerick,

Thank you for submitting your manuscript to PLOS ONE. After careful consideration, we feel that it has merit but does not fully meet PLOS ONE’s publication criteria as it currently stands. Therefore, we invite you to submit a revised version of the manuscript that addresses the points raised during the review process.

We look forward to receiving your revised manuscript.

Kind regards,

Jun Hyeok Lim, M.D.

Academic Editor

PLOS ONE

Reviewers' comments:

Reviewer's Responses to Questions

**Comments to the Author**

1. Is the manuscript technically sound, and do the data support the conclusions?

Reviewer #1: Yes

Reviewer #2: Yes

2. Has the statistical analysis been performed appropriately and rigorously? 

Reviewer #1: Yes

Reviewer #2: N/A

3. Have the authors made all data underlying the findings in their manuscript fully available?

Reviewer #1: No

Reviewer #2: Yes

4. Is the manuscript presented in an intelligible fashion and written in standard English?

Reviewer #1: Yes

Reviewer #2: No

5. Review Comments to the Author

Reviewer #1: Dear Author,

The manuscript presents a thorough investigation into the effects of low-dose computed tomography (LDCT) screening for lung cancer in regions with a high prevalence of tuberculosis (TB). The systematic review and meta-analysis approach are commendable and provide valuable insights into this important topic.

The methodology section is well-structured, and the use of the PICOS framework for defining eligibility criteria adds clarity to the research process. Adherence to PRISMA guidelines enhances the transparency and reproducibility of the study.

The statistical analysis appears to have been conducted appropriately, supporting the conclusions drawn by the authors. However, the lack of full availability of underlying data is noted, which may impact the reproducibility of the findings. It would be beneficial to address this limitation to strengthen the credibility of the research.

Overall, the manuscript is presented clearly and written in standard English, making it accessible to readers. The study contributes valuable insights into the effectiveness of LDCT screening in TB-endemic regions. Addressing the availability of underlying data would further enhance the quality and impact of the research.

Reviewer #2: native English needs to be reviewed

small sample size needs to be widen the sample in order to fundamentalize the conclusions

It would be change some guidelines in the follow up and interpretation of cancers and TB

6. PLOS authors have the option to publish the peer review history of their article (what does this mean?). If published, this will include your full peer review and any attached files.

Reviewer #1: No

Reviewer #2: **Yes: **Marwa I .Khalaf

---

## [Author Response · Author response to Decision Letter 0]

2 May 2024

Dear Editor-in-Chief of Plos One,

Thank you for the meticulous review of the manuscript. Below are the comments and answers for the manuscript submitted to PLOS ONE under the number “PONE-D-23-38324”.

Kind Regards,

Isabel Emmerick

Reviewer’s Responses to Questions

Comments to the Author

1. Is the manuscript technically sound, and does the data support the conclusions?

Reviewer #1: Yes

Reviewer #2: Yes

2. Has the statistical analysis been performed appropriately and rigorously?

Reviewer #1: Yes

Reviewer #2: N/A

3. Have the authors made all data underlying the findings in their manuscript fully available?

The PLOS Data policy<http://www.plosone.org/static/policies.action#sharing> requires authors to make all data underlying the findings described in their manuscript fully available without restriction, with rare exception (please refer to the Data Availability Statement in the manuscript PDF file). The data should be provided as part of the manuscript or its supporting information or deposited to a public repository. For example, in addition to summary statistics, the data points behind means, medians, and variance measures should be available. If there are restrictions on publicly sharing data—e.g. participant privacy or use of data from a third party—those must be specified.

Reviewer #1: No

Reviewer #2: Yes

Authors’ answers: All the data used to prepare the graphs and analysis are available in the body of the manuscript within the tables. Due to copyright restrictions, the original manuscripts can not be shared. Nonetheless, they are all appropriately cited in the paper and can be found online.

4. Is the manuscript presented in an intelligible fashion and written in standard English?

Reviewer #1: Yes

Reviewer #2: No

Authors’ answers: An english review was perfomed

5. Review Comments to the Author

Reviewer #1: Dear Author,

The manuscript presents a thorough investigation into the effects of low-dose computed tomography (LDCT) screening for lung cancer in regions with a high prevalence of tuberculosis (TB). The systematic review and meta-analysis approach are commendable and provide valuable insights into this important topic.

The methodology section is well-structured, and the use of the PICOS framework for defining eligibility criteria adds clarity to the research process. Adherence to PRISMA guidelines enhances the transparency and reproducibility of the study.

The statistical analysis appears to have been conducted appropriately, supporting the conclusions drawn by the authors. However, the lack of full availability of underlying data is noted, which may impact the reproducibility of the findings. It would be beneficial to address this limitation to strengthen the credibility of the research.

Overall, the manuscript is presented clearly and written in standard English, making it accessible to readers. The study contributes valuable insights into the effectiveness of LDCT screening in TB-endemic regions. Addressing the availability of underlying data would further enhance the quality and impact of the research.

Authors’ answers: Thank you for your comments. The purpose of meta-analysis is not to reanalyze the original data but to aggregate and synthesize the results reported in the studies included in the analysis. Meta-analysis provides a quantitative estimate or summary of the findings from multiple studies on a particular topic, allowing researchers to draw more robust conclusions than those based on individual studies alone. By pooling data from various sources, meta-analysis can identify patterns, trends, and overall effects that may not be apparent in single studies. Therefore, the focus is on presenting and interpreting the collective evidence. In addition, we systematically evaluate the quality of the papers included and the risk of bias. All summarized data is available within the manuscript tables, allowing the reproducibility of the findings.

Additionally, we included a phrase on not performing a re-analysis of the original data in the limitation section.

Reviewer #2: native English needs to be reviewed

small sample size needs to be widen the sample in order to fundamentalize the conclusions

It would be change some guidelines in the follow up and interpretation of cancers and TB

Authors’ answers: Thank you for your comments. 

1) The English was revised prior to this version submission.

2) “small sample size needs to be widen the sample in order to fundamentalize the conclusions” 

In our systematic review, we followed a specific protocol registered in the PROSPERO database (registry number CRD42022309581). We included all the available evidence that met the inclusion criteria and adhered to recommended quality international guidelines. Therefore, the term “sample size” is not applicable in systematic reviews, which are exhaustive by definition.

3) It would be change some guidelines in the follow up and interpretation of cancers and TB - About the guidelines in the usage of radiological investigations in TB and their narrow interpretation so to be following as much as possible the basic guidelines and present them in a clear manner.

We included the following text in the manuscript: “Insights from this literature review may inform future guidelines on delivering LDCT screening in low- and middle-income countries (LMICs) with high TB burden. Our results emphasize the importance of implementing structured screening programs.”

---

## [Editor Report · Decision Letter 1]

17 Jul 2024

Effects of Low Dose Computed Tomography (LDCT) on lung cancer screening on incidence and mortality in regions with high tuberculosis prevalence: a systematic review

PONE-D-23-38324R1

Dear Dr. Emmerick,

We’re pleased to inform you that your manuscript has been judged scientifically suitable for publication and will be formally accepted for publication once it meets all outstanding technical requirements.

Kind regards,

Yuchen Qiu, Ph.D.

Academic Editor

PLOS ONE
---

## [Editor Report · Acceptance letter]

2 Sep 2024

PONE-D-23-38324R1 

PLOS ONE

Dear Dr. Emmerick, 

I'm pleased to inform you that your manuscript has been deemed suitable for publication in PLOS ONE. Congratulations! Your manuscript is now being handed over to our production team.

Kind regards, 

on behalf of

Dr. Yuchen Qiu 

Academic Editor

PLOS ONE